

# Current observed global mean sea level rise and acceleration estimated from satellite altimetry and the associated uncertainty

Adrien Guérou[1], Benoit Meyssignac[2,3], Pierre Prandi[1], Michaël Ablain[4], Aurélien Ribes[5], and François Bignalet-Cazalet[3]

[1]Collecte Localisation Satellite (CLS), Ramonville Saint-Agne,31250, France
[2]LEGOS, CNRS, IRD, Université Paul Sabatier, Toulouse, 31400, France
[3]Centre National d'Etudes Spatiales (CNES), 31400 Toulouse, France
[4]MAGELLIUM, Ramonville Saint-Agne, 31520, France
[5]CNRM, Université Paul Sabatier, Météo France, CNRS, Toulouse, 31400, France

**Correspondence:** Adrien Guérou (aguerou@groupcls.com)

**Abstract.** We present the latest released of the Global Mean Sea Level (GMSL) record produced by the French space agency CNES and distributed on the AVISO+ website. This dataset is based on reprocessed along-track data, so-called L2P 21, of the reference missions Topex-Poseïdon, Jason-1/-2 and -3. The L2P 21 CNES/AVISO GMSL record covers the period January-1993 to December-2021 and is now delivered with an estimate of its uncertainties following the method presented in Ablain et al. (2019). Based on the latest Calibration and Validation (Cal/Val) knowledge, we updated the uncertainty budget of the reference altimetry missions and demonstrate that the CNES/AVISO GMSL record now achieves stability performances of $\pm 0.3\,mm/yr$ at the 90% confidence level (C. L.) for its trend and $\pm 0.05\,mm/yr^2$ (90%C. L.) for its acceleration over the 29-years of the altimetry record. Thanks to an analysis of the relative contribution of each uncertainty budget contributor, i.e., the altimeter, the radiometer, the orbit determination, the geophysical corrections, we identified the current limiting factors to the GMSL monitoring stability and accuracy. We find that the radiometer Wet Troposphere Correction (WTC) and the high-frequency errors with timescales shorter than 1-year are the major contributors to the GMSL uncertainty over periods of 10-years (30-70%), both for the trend and acceleration estimations. For longer periods of 20-years, the TP data quality is still a limitation but more interestingly, the International Terrestrial Reference Frame (ITRF) realisation uncertainties becomes dominant over all the others sources of uncertainty. Such a finding challenges the altimetry observing system as it is designed today and highlights clear topics of research to be explored in the future to help the altimetry community to improve the GMSL accuracy and stability.

## 1  Introduction

Since october 1992 and the launch of Topex-Poseïdon, radar satellite altimetry has proved its capacity to monitor the small sea level variations induced by the natural climate variability and by the anthropogenic climate change (e.g. Cazenave and Moreira 2022). An important effort of space agencies and the sea level science community over the last 30 years has continuously improved the satellite altimetry observing system and the associated data record leading to a sea level record with unprecedented



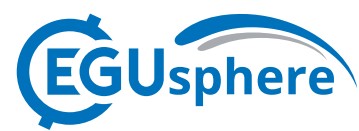

coverage, resolution, accuracy and stability. Now, sea level is currently monitored on a daily basis, from 82°S to 82°N, with a 1/4°x1/4° resolution, with an accuracy better than $\pm 1.5\ cm$ on 1 Hz measurements and a stability below $\pm 0.5\ mm/yr$ on decadal time scales. This accuracy and stability enable to detect, analyse and attribute sea level changes in response to the

climate variability at global and regional scale. As time is passing and the sea level signal that is forced by greenhouse gazes (GHG) emissions is unravelling above the internal climate variability, the sea level record also becomes a reference to assess and validate climate model simulations of sea level change in response to GHG emissions (e.g., Nerem et al., 2018). Sea level is also a key variable to constraint ocean reanalysis as it provides a unique estimate of the geostophic circulation over the whole water column which is a central piece of information to determine the general ocean circulation. For all these reasons the sea

level record retrieved by satellite altimetry has become a reference for scientists to study ocean circulation changes and climate changes and for stakeholders to support their decisions and strategies on adaptation to climate-change induced sea level rise (ref IPCC SROCC chap 4, IPCC AR6 chap 9).

To ensure the best possible estimate of current sea level changes, space agencies regularly revisit and update the production

of the sea level record from the satellite archive. In addition, since 2009, they also provide careful estimates of the associated uncertainty to deliver to users information on the reliability and the accuracy of the sea level estimates.

This work presents the new release of the global mean sea level record (GMSL) and its associated uncertainty from the AVISO project of the Centre National d'Etudes Spatiales (CNES).

First we present the satellite altimetry data that is used (section 2.1).Then we recall how the GMSL is derived and we detail the new updates in the GMSL computation since the last release (section 2.2). In section 2.3 we update the GMSL error budget and the GMSL error variance/covariance matrix. We derive in section 3 the estimates of the GMSL anomalies, trends and accelerations since 1992 with their associated uncertainty. We explore in section 4 the sources of uncertainty in the GMSL trends and accelerations on different time scales. We identify the correlated noise at 1yr, the Wet Troposphere Correction

(WTC) and the International Terrestrial Reference Frame (ITRF) realisation as the major sources of uncertainty on time scales longer than 10 years. On this basis we propose in section 5 directions of research to reduce the uncertainties in the GMSL trends and accelerations.

## 2  Data and methods

### 2.1  Altimetry dataset

The CNES/AVISO GMSL record is computed based on the Level-2 + (L2P) CNES/AVISO 1 Hz Non Time Critical (NTC) along-track data of the altimetry reference missions Topex-Poseïdon (TP), Jason-1 (J1), Jason-2 (J2) and Jason-3 (J3). The latest reprocessing of these products is used, i.e. , version V03_00, hereinafter referred as L2P 21. The L2P 21 products benefit from reprocessed data of individual missions and homogeneous state-of-the-art geophysical corrections to ensure accurate and stable climate data records (see Legeais et al., 2021, for a comprehensive description of the altimetry dataset production within





**Table 1.** Origins and references of the corrections contained in the L2P 21 along-track 1 Hz products. These products are the ones used to compute the current GMSL CNES/AVISO record. The terms in blue are the ones updated as compare to the previous version L2P 18.

| Geophysical correction | Topex-Poseïdon | Jason-1 | Jason-2 | Jason-3 |
|---|---|---|---|---|
| Orbit | GDR GSFC STD18 | CNES POE-E | CNES POE-F | |
| Range | M-GDR | GDR-E | GDR-D | GDR-D/-F |
| Sea State Bias | CLS update (Tran et al., 2010) | GDR-E | CLS update (Tran et al., 2012) | CLS update (based on Tran et al., 2012) |
| Ionosphere | CLS update (Nencioli, 2021) | | | |
| Wet Troposphere | CLS update (Fernandes and Lázaro, 2016) | GDR-E (from radiometer) | GDR-D (from radiometer) | GDR-D/-F (from radiometer) |
| Dry Troposphere | CLS update (from ERA-5 sea level pressure model) | | | |
| DAC | CLS update From ERA-5 model (Carrere et al., 2020) | | | GDR-D/-F (Carrère and Lyard, 2003) |
| Ocean tide | CLS update FES2014 (Carrere et al., 2014; Lyard et al., 2021) | | | GDR-F FES2014 |
| Internal tide | CLS update (Zaron, 2019; Carrere et al., 2021) | | | |
| Solid Earth tide | GDR (Cartwright and Tayler, 1971; Cartwright and Edden, 1973) | | | |
| Pole tide | CLS update (Desai et al., 2015) | | | |
| MSS | CLS update (composite SCRIPPS, CNES/CLS 15, DTU 15, see DOI) | | | |

the Copernicus program). A complete description of the L2P 21 dataset is given in the "Along-track Level-2 + (L2P) SLA products" handbook -link

      The main improvements brought by the L2P 21 standards, as compared to those of the previous version (Ablain et al., 2016, 2019) are summarized hereafter : a new Dynamical Atmospheric Correction (DAC) solution is used that is based on ERA 5 data and computed with the TUGO model (Carrere et al., 2020). It yields to a reduction of the Sea Surface Height (SSH)

variance at crossover points of 5%. A combined Mean Sea Surface (MSS) is also used, now computed with respect to the World Geodetic System (WGS) 84 reference ellipsoid, instead of the TP ellipsoid. It conducts to a better stability of the Sea Level Anomaly (SLA) variable, especially at high latitudes. The pole tide solution has been improved with a better definition of its mean location (Desai et al., 2015; Ries and Desai, 2017) and an internal tide solution has been added to the calculation



of the SLA (Zaron, 2019). Finally, Jason-2 and Jason-3 missions now benefit from the use of their official Geophysical Data
Record (GDR) Wet Troposphere Correction (WTC) from their respective on-board radiometer instruments.

A summary of the altimetry variables and geophysical corrections contained in the L2P 21 products of the four reference
missions is presented in Table 1. Note that the on-going GDR - F reprocessing of the Topex-Poseïdon data has not been included
in the L2P 21 and that Jason-3 GDR - F data is only used from cycle 171 on-wards (October 2020).

## 2.2  GMSL computation

The L2P 21 GMSL record has been computed following the AVISO method (Henry et al., 2014, section 2.1). In a nutshell,
the along-track 1 Hz SLA measurements are first averaged within grid-cells of $1 \times 3$ degrees for each orbital cycle (∼10 days).
Then, all grid cells within $\pm 66°$ N/S are spatially averaged for each cycle, with a weighting that accounts for the relative ocean
area covered. This gives one GMSL measurement in time. In practice, the weights are a function of the cosine of the latitude
and of the ocean-to-land ratio of each grid-cell. As compared to Henry et al. (2014), the AVISO method now uses grid-cell size
of $1 \times 3$ degrees in latitude and longitude, respectively. Grid-cells around the tropics and coastlines are thus better populated to
mitigate trends overestimation in these regions, as noted in Henry et al. (2014, see also this AVISO note -link) and confirmed
by Scharffenberg and Stammer (2019).

### 2.2.1  GMSL intermissions offsets

The L2P 21 GMSL is currently built from four altimetry missions : Topex-Poseïdon, Jason-1, Jason-2 and Jason-3. These
missions flew successively on the same orbit since 1992 with calibration phases, called "tandem phases", during which the
successive satellites fly less-than-a-minute apart over the same ground-track. During tandem phases, consecutive missions
observe precisely the same sea level such that the GMSL intermission offset (due to non-correlated instrumental differences)
can be accurately estimated, and corrected for. These tandem phases generally last from 6 to 12 months and are key to ensure
the long-term continuity and stability of the GMSL record (Dorandeu et al., 2004; Leuliette et al., 2004; Zawadzki and Ablain,
85  2016).

In practice, the global intermissions offsets are computed as the mean difference of the respective GMSL values over a given
sub-period (i.e. , a given number of cycles) of the tandem phase. In the previous version of the CNES/AVISO GMSL record,
only nine cycles within each tandem phases (out of about twenty) centered around the switching date from one mission to
another, were used to compute the intermissions offsets. This was different from other groups who used the whole tandem
phases (Masters et al., 2012; Henry et al., 2014). Based on an improved estimation method of the GMSL intermissions offsets
uncertainties (detailed hereafter) we show that uncertainties are reduced when using as many measurements as possible. For
this reason, the GMSL intermission offsets of the L2P 21 GMSL are now computed based on all available tandem phase cycles.
Table 2 summarises the different missions we used and the respective periods over which they were used.

Figure 1 shows the GMSL timeseries of the four reference missions with a focus on their tandem phases. Bottom panels of
Figure 1 show that the difference of the GMSL timeseries over the tandem phases can be approximated, at first order, by a
constant. For a specific mission switch, i.e. , between J2 and J3 for instance, the intermission offset is thus simply estimated as



**Table 2.** Altimetry missions used to establish the CNES/AVISO GMSL record. The periods covered by each mission in the GMSL record are provided in the second column, the corresponding cycles are given in column three and the tandem phases cycles used to compute the intermissions offsets are given in the last three columns.

| Mission | Period (start/end) | Cycle number | Tandem phase | | |
|---|---|---|---|---|---|
| - | [Y-m-d] | - | [cycle] | | |
| | | | *TP/J1* | *J1/J2* | *J2/J3* |
| Topex-A/B | 1992-12-31 *to* 2002-05-04 | 11 - 354 | 344 - 364 | | |
| Jason-1 | 2002-05-04 *to* 2008-10-29 | 12 - 250 | 1 - 21 | 240 - 259 | |
| Jason-2 | 2008-10-29 *to* 2016-06-05 | 12 - 291 | | 1 - 20 | 281 - 303 |
| Jason-3 | 2016-06-05 *to* 2021-10-20 | 12 - 208 | | | 1 - 23 |

**Figure 1.** GMSL record of the reference missions with a focus on the respective tandem phases. Top panel indicates the position in time of the respective tandem phases. Middle panels show zooms on the GMSL records over the tandem phases and bottom panels show the GMSL record differences between the two respective missions in tandem phase (the mean value of the timeseries have been removed to ease the comparison between tandem phases).

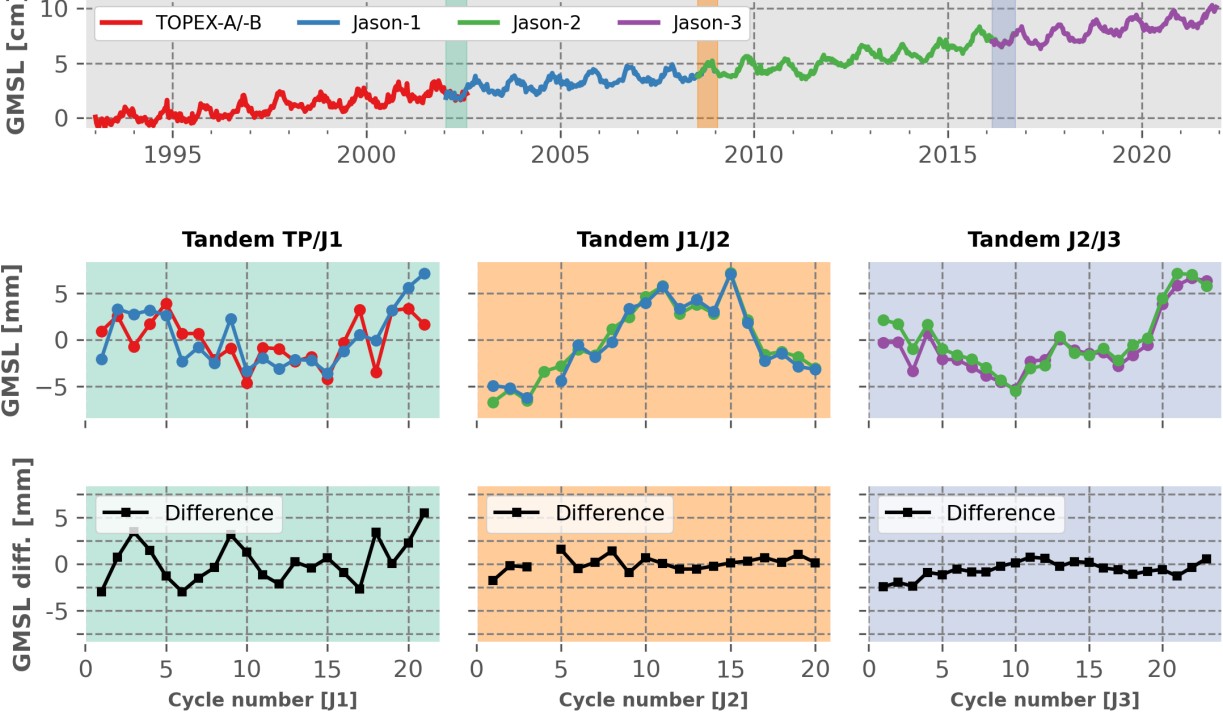





the mean of the GMSL differences over all the tandem phase cycles, i.e., the mean of the black curve on the bottom-right panel of Figure 1. As a consequence, statistically speaking, the uncertainty associated to this mean calculation is the uncertainty associated to the estimation of the mean of a population, when only a sample of this population is known. In this situation,

where the original population variance is unknown (we do not have more measurements than the tandem phase samples), the Student law can be applied to describe the population's mean, such that:

$$\bar{X} = \bar{x} \pm t_{1-\alpha/2}^{n-1} \frac{s}{\sqrt{n}} \tag{1}$$

with, $\bar{X}$ the mean of the population, $\bar{x}$ the mean of the sample, $n$ the number of independent measurements of the sample, $t_{1-\alpha/2}^{n-1}$ the Student coefficient for $n-1$ degree of freedom and a confidence level of $\alpha$, and $s$ the standard deviation of the

sample. The uncertainty of the mean estimation is thus given by the second member of equation (1), which is in our case the GMSL intermission offset uncertainty we look for. It is important to note that for low values of $n$ (typically lower than 30) equation (1) is valid if and only if: (a) the sample population follows a normal law, (b) the measurements are independent from each other.

We performed Shapiro-Wilk tests for the three GMSL difference timeseries over the tandem phases and confirm that their

distribution is not clearly non-Gaussian (i.e., p-values greater than 0.05). To derive the number of independent measurements, we assumed that all the GMSL difference timeseries follow auto-regressive processes of the first order (AR1) and thus, the number of independent measurements can be estimated using the following equation (von Storch and Zwiers, 1999):

$$n = (1 - \rho_1)/(1 + \rho_1) \times n_{sample} \tag{2}$$

with, $\rho_1$ the auto-correlation of the population at lag-1 and $n_{sample}$ the total number of measurements of the sample. We

note that assuming an AR1 process might overestimate the number of independent points and thus underestimate the offset uncertainties in the case where the processes would be of higher orders.

From equation (1) and (2) we thus estimated the GMSL intermission offset uncertainties for the three tandem phases of the reference missions considering a varying number of cycles within each tandem phase. We found that the uncertainties are lowest when using all cycles available because the number of independent point increases. Resulting values are summarised in

Table 3, along with their statistical characteristics.

The obtained offset uncertainties are lower than the ones from the previous CNES/AVISO GMSL record, i.e., 0.5 mm at $1\sigma$ (Zawadzki and Ablain, 2016), mainly due to the use of the whole tandem phase. We thus provide the L2P 21 GMSL record with corrected and adapted intermissions offsets and uncertainties for each tandem phases. Note that the offset between the TP-A altimeter and the redundant TP-B altimeter has not bee revisited. Values from the previous version of the CNES/AVISO

GMSL are used, that are based on Ablain et al. (2019).



**Table 3.** L2P 21 GMSL intermissions offset uncertainties ($1\sigma$) as estimated in this paper. The results (bottom line) can be re-estimated with eq. (1) and the different parameters given here.

| Tandem phase | TP / J1 | J1 / J2 | J2 / J3 |
|---|---|---|---|
| Shapiro-Wilk p-value | 0.52 | 0.82 | 0.31 |
| Degree of freedom $n$ | 14 | 14 | 6 |
| Standard deviation $s$ [mm] | 2.26 | 0.78 | 0.85 |
| **Offset uncertainty ($1\sigma$) [mm]** | **0.3** | **0.1** | **0.2** |

### 2.2.2 GMSL global corrections

The reference GMSL record available on the AVISO+ website is provided with different optional global corrections that the user may use according to its need.

First, we provide for the first time the empirical correction to account for the TP‑A altimeter drift well documented by the community (Valladeau et al., 2012; Watson et al., 2015; Dieng et al., 2017; Beckley et al., 2017; Cazenave et al., 2018). We use the empirical correction proposed by Ablain et al. (2017) that can be approximated by a "V-shape" of $-1\,mm.yr^{-1}$ over the January, 1993 and July, 1995 period, and $+3\,mm.yr^{-1}$ over the August, 1995 and February, 1999 period. The associated uncertainties are described in Table 4. Second, a Global Isostatic Adjustement (GIA) correction of $+0.3\,mm.yr^{-1}$ is available over the entire data record to account for the post glacial rebound of the earth crust (Spada, 2017). Finally, we warn the users that an additional correction of $+0.1\,mm.yr^{-1}$ need to be applied on the L2P 21 GMSL record when compared to GRACE and/or ARGO data to account for the deformation of the ocean bottom (Frederikse et al., 2017; Lickley et al., 2018). This correction is not distributed on the AVISO+ website.

For the rest of the paper, we use the reference L2P 21 GMSL record corrected for the TP‑A altimeter drift as well as for the GIA correction. The resulting record covers the period from January 1993 to December 2021, hence providing a $\sim$ 29-year data record of the GMSL. The dataset is available to download at the following address: www.aviso.altimetry.fr/gmsl/data_access.

### 2.3 GMSL uncertainty budget

The CNES/AVISO GMSL record is delivered with an updated estimate of its uncertainties following the method developed in Ablain et al. (2009) and extended in Ablain et al. (2019). This method is based on the construction of an uncertainty budget, as comprehensive as possible, of the GMSL timeseries itself. The uncertainty budget includes a description of the time covariance of the errors which allows to quantify the uncertainty envelop of the GMSL timeseries as well as to estimate consistently its trend and acceleration uncertainties (see Sect. 2.4). Limitations of such approach is discussed in Section 5.

The main differences between the L2P 21 and L2P 18 GMSL version come from the use of new geophysical corrections (i.e. , DAC, internal tide, MSS) as well as the use of the WTC from the on-board radiometer instruments of J2 and J3 missions (see Sect. 2.1). No new altimetry missions, neither reprocessing of their altimeter data has been used. Therefore, the sources





of uncertainties that affect the L2P 21 GMSL record are the same as the ones presented in Ablain et al. (2019). The exact list is provided in Table 4. We recall that all the uncertainties considered are assumed to follow a Gaussian distribution and are independent from one another.

However, the level of uncertainty for a few sources of uncertainties need to be updated and adapted to the new L2P 21 GMSL
record. This is the case of the correlated uncertainties at timescale of 2-months and 1-year which are estimated empirically directly from a filtering of the GMSL timeseries (Ablain et al., 2019, section 3). We updated these sources of uncertainties by filtering the L2P 21 GMSL record in the same manner and obtained the values presented in the first two rows of Table 4. As compare to the L2P 18 GMSL, the correlated uncertainties at timescale of 2-months and 1-year of the L2P 21 record are of the same order of magnitude and/or lower. This is mainly due to the improvements of the geophysical corrections contained
in the L2P 21 along-track products (see Section 2.1). It is important to recall here that this method tends to overestimate the true uncertainties of the GMSL record since it does not exclude any physical signals, i.e. , some geophysical signal might be considered as noise.

We also updated the uncertainty associated to the WTC over the Jason-3 period. Recent work by Barnoud et al. (2021) highlighted a potential long-term drift of the Jason-3 radiometer measurements when compared to Saral/AltiKa and Sentinel-
3A radiometers at cross-over points. The drifts observed are systematically positive (respectively $0.8$ and $0.5\,mm.yr^{-1}$) and larger than the typical radiometers drift uncertainties expected from altimetry missions, i.e. , $\pm 0.2\,mm.yr^{-1}$ over 5-years period (Ablain et al., 2009; Thao et al., 2014; Legeais et al., 2014). We thus decided to update the uncertainty associated to the WTC over the Jason-3 period. We took the most conservative value considering that the drifts observed in Barnoud et al. (2021) are equally due to any radiometers on-board of the three altimetry missions considered. As detailed in Table 4, we took
a value of $0.8\,mm.yr^{-1}/\sqrt{2} = 0.55\,mm.yr^{-1}$ to quantify the WTC uncertainty over J3 period. In practice, we prescribed a correlated error at 5-years timescale that creates the same drift uncertainty as a drift error over a 5-years period (as in Ablain et al. 2019).

Finally, we updated the uncertainties associated to the GMSL intermissions offsets as presented in details in Section 2.2.1. As for the intermission offset uncertainty between TP-A and the redundant altimeter TP-B, we use the value of 2 mm at $1\sigma$
from Ablain et al. (2019), see Table 4.

## 2.4 Estimation method of the GMSL trend, acceleration and uncertainties

To estimate the sea level rise and acceleration of the L2P 21 GMSL record, we first apply a 2-months low-pass Lanczos filter to the timeseries. We then fit a quadratic regression model to the filtered timeseries following an Ordinary Least Square (OLS)
approach, as described in Ablain et al. (2019, section 6). This regression model also contains semi-annual and annual signals that are adjusted simultaneously (i.e. , amplitudes and phases) through the OLS estimator. We use the GMSL uncertainty budget established in Sect. 2.3 to construct a variance-covariance matrix (Ablain et al., 2019, section 4) that we use, through the OLS estimator, to estimate the uncertainty envelop of the GMSL record, as well as its trend and acceleration uncertainties. Figure 2 shows the variance-covariance matrix of the L2P 21 GMSL record.





**Table 4.** L2P 21 GMSL uncertainty budget given at $1\sigma$. The sources of uncertainties are based on the work of Ablain et al. (2019) and have not been changed. The values in blue are the one updated as compared to the uncertainty budget of the previous CNES/AVISO GMSL record.

| Source of uncertainties | Type of errors | Uncertainty ($1\sigma$) | Method / References |
|---|---|---|---|
| Altimeter noise / geophysical corrections | Correlated errors $\lambda =$ 2-months | $u_\sigma = 1.7\,\text{mm}$ over TP period<br>$u_\sigma = 1.2\,\text{mm}$ over J1 period<br>$u_\sigma = 1.1\,\text{mm}$ over J2 period<br>$u_\sigma = 1.0\,\text{mm}$ over J3 period | This paper (Sect. 2.3) |
| Geophysical corrections / orbit | Correlated errors $\lambda =$ 1-year | $u_\sigma = 1.4\,\text{mm}$ over TP period<br>$u_\sigma = 1.2\,\text{mm}$ over J1 period<br>$u_\sigma = 1.1\,\text{mm}$ over J2 period<br>$u_\sigma = 1.1\,\text{mm}$ over J3 period | This paper (Sect. 2.3) |
| Radiometer WTC | Correlated errors $\lambda =$ 5-years | $u_\sigma = 1.1$ mm over TP, J1, J2 periods<br><br>$u_\sigma = 1.8\,\text{mm}$ over J3 period | Legeais et al. (2014)<br>Thao et al. (2014)<br>This paper (Sect. 2.3) |
| Orbits determination | Correlated errors $\lambda =$ 10-years | $u_\sigma = 1.12$ mm over TP period<br>$u_\sigma = 0.5$ mm over Jasons period | Couhert et al. (2015);<br>Rudenko et al. (2017) |
| Intermissions calibration offsets | Bias | $u_\Delta = 2\,mm$ for TP-A/B<br>$u_\Delta = 0.3\,mm$ for TP/J1<br>$u_\Delta = 0.1\,mm$ for J1/J2<br>$u_\Delta = 0.2\,mm$ for J2/J3 | This paper (sec. 2.2.1) |
| International Terrestrial Reference Frame (ITRF) | Drift | $u_\delta = 0.1\,mm/yr$ over 1993-present | Couhert et al. (2015) |
| Global Isostatic Adjustement (GIA) | Drift | $u_\delta = 0.05\,mm/yr$ over 1993-present | Spada (2017) |
| Topex-A/-B altimeter drift | Drift | $u_\delta = 0.7\,mm/yr$ over TP-A period<br>$u_\delta = 0.1\,mm/yr$ over TP-B period | Ablain et al. (2017) |

We recall from Ablain et al. (2019) that the main advantages of using such an OLS estimator for climate variables are: "(i) OLS is consistent with previous estimators of GMSL trends as well as estimators of trends in other essential climate variables than GMSL [..] and that (ii) the OLS best estimate does not depend on the estimated variance–covariance matrix". This also means that the uncertainty estimates only depend on the variance-covariance matrix construction.

## 3    Results

This section presents the L2P 21 CNES/AVISO GMSL record, its trend and acceleration estimates along with their uncertainties. The analysis is based on the data detailed in Sect. 2.2 that we filtered with a 2-months low-pass filter and from which we



**Figure 2.** Error variance-covariance matrix of the L2P 21 GMSL obtained from the revised uncertainty budget presented in Table 4

removed the semi-annual and annual components (see Sect. 2.4). We recall that corrections for the GIA and TP-A drift have also been applied (see Sect. 2.2.2).

## 3.1 GMSL record and its uncertainty envelope

Figure 3 shows the L2P 21 GMSL record (red curve) along with its 90% C. L. uncertainty envelop. The uncertainty envelop is derived by taking the square roots of the diagonal terms of the variance-covariance matrice (see Figure 2) obtained from the uncertainty budget presented in Sect. 2.3. A direct comparison to the former L2P 18 version is shown on the top panel of Figure 3. Differences of the order of $1 - 2\,mm$ can be observed, with an annual and semi-annual signal pattern, mainly due to the changes of geophysical standards (see Sect. 2.1). The larger differences over the TP period ($\sim 2\,mm$) are consistent with

the larger variability of the TP record (see the high frequency errors Table 4). A peak of $\sim 4\,mm$ is also visible in 2013 and





corresponds to cycle-174 of Jason-2, where a large amount of tracks are flagged as invalid in both GMSL record versions. The GMSL value of this cycle is thus highly sensitive to change in the geophysical standards as the number of points used to get the GMSL average is low. All these differences are well within the 90 % C. L. uncertainty envelop and do not generate any significant changes in decadal GMSL trends and accelerations (see also the following Sect. 3.2).

In Figure 4, we show the uncertainty envelopes at the 90 % C. L. level for both the L2P 21 and L2P 18 GMSL records. The general shape of the uncertainty envelop is a parabola centered around the central time of the data record, with largest uncertainties during TP period ($4 - 8\,mm$), smallest uncertainties during J1/J2 period ($3 - 4\,mm$) and increasing uncertainties at the end of the data record (J3, $4 - 5\,mm$). We recall from Ablain et al. (2019) that we obtain minimum GMSL uncertainties in the central period of the timeseries as it benefits from prior and posterior measurements. It actually corresponds to the date

when the errors and their cumulative time correlation is the smallest.

Compared to the L2P 18 uncertainty envelope, Figure 4 shows that the uncertainty envelop of the L2P 21 GMSL record is smaller by $\sim 0.5\,mm$ during J1 period and by $\sim 0.2\,mm$ during J2/J3 periods. This is a direct consequence of the lower uncertainties correlated at 2-months and 1-year measured on the L2P 21 GMSL for these missions (see Table 4). We also note a larger increase of the uncertainty envelope over J3 period that corresponds to the larger uncertainty of its radiometer WTC (see

Sect. 2.3 and Table 4). Finally, we observe changes in the uncertainty envelop at the connection between missions between the two version of the GMSL records. This is due to a combination of the (i) improvements of the intermissions offset uncertainties, (ii) smaller/larger uncertainty differences between two consecutive missions for the correlated errors at 2-months and 1-year.

## 3.2   GMSL rise, acceleration and uncertainties

As shown in Figure 3, we estimate the GMSL rise over the 29-years of the altimetry record, from January-1993 to December-

2021, to $3.3 \pm 0.3\ mm.yr^{-1}$. We estimate the acceleration of the GMSL rise to $0.12 \pm 0.05\ mm.yr^{-2}$ which confirms previous analysis in the literature demonstrating that the GMSL record is accelerating (Watson et al., 2015; Dieng et al., 2017; Beckley et al., 2017; Nerem et al., 2018; Ablain et al., 2019; Veng and Andersen, 2020). The trend and acceleration estimates of the previous L2P 18 GMSL record, over the period January-1993 to October-2017, also corrected for the TP-A drift, were $3.1 \pm 0.4\ mm.yr^{-1}$ and $0.12 \pm 0.07\ mm.yr^{-2}$, respectively (Ablain et al., 2019)

Based on the approach described in Sect. 2.4, we estimated the GMSL rise and acceleration, as well as their uncertainties, for any time span between 1-year and 29-years included in the period covered by the data. The results are shown in Figure 5 for the trends and in Figure 6 for the accelerations. The top points of the triangles are thus corresponding to the total length of the data record. Uncertainty values are given at the 90 % C. L. and only the respective significant values of sea level rise and/or acceleration are shown (e.g. , larger than the uncertainties).

Figure 5 shows that the GMSL rise is significant at the 90 % C. L. for any periods longer than 5 years. For shorter periods, the GMSL trends are mostly lower than their respective uncertainties, i.e. , $1 - 2\ mm.yr^{-1}$. For periods longer than 5-years, the GMSL rise ranges from $2.5$ to $5.5\ mm.yr^{-1}$ with the largest values centered around 2014 for periods of $5 - 8$ years. Regarding the GMSL trend uncertainties, the diagram shows a similar pattern as in Ablain et al. (2019) : the longer the period, the smaller the trend uncertainty, with a steep inflection of the iso-uncertainty lines as the central year gets towards the TP period. This





**Figure 3.** (Bottom panel) CNES/AVISO L2P 21 GMSL record and its associated uncertainties. The record has been corrected for TP-A drift as well as for the GIA. Seasonal signals are removed and the timeseries is 2-monhts filtered. The uncertainty envelop (red shaded area) is given at the 90% C. L. (1.65σ), as for the trend and acceleration uncertainties indicated in the white box. (Top panel) Comparison of the L2P 21 GMSL record with its previous version, i.e., L2P 18. Vertical dashed-lines indicated the dates of switch between altimetry missions.

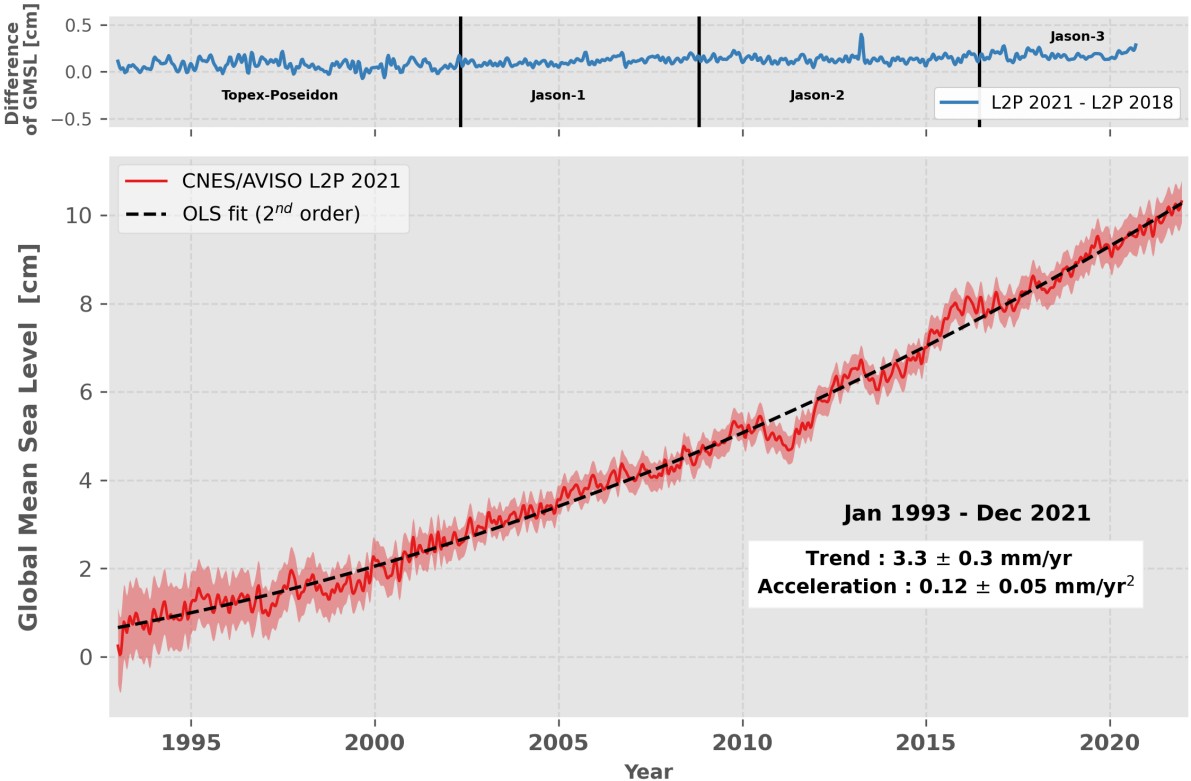

behaviour is due to the large contributions of the TP drift uncertainties and the intermission offsets uncertainty between TP-A and TP-B to the total GMSL uncertainty budget (see Appendix A and Sect. 5 for more details). We also observe a small inflection of the iso-uncertainty lines from 2015-onwards that corresponds to the increase of the uncertainty of the WTC of J3 as compared to other missions. It is interesting to note that the minimum GMSL trend uncertainty is less than $0.3\ mm.yr^{-1}$. It is reached for the 24-years period centered in 2010.

On the acceleration, we observe on Figure 6 that the GMSL acceleration is significant only for some periods longer than 10 years, mostly centered around 2011. When significant, the acceleration ranges from $0.12 - 0.60\ mm.yr^{-2}$. The largest values are reached for the smallest period considered (i.e., 10-years centered around 2011) suggesting that these large acceleration values are caused by the internal variability of the climate system. The uncertainties of the GMSL acceleration ranges from $0.05 - 0.35\ mm.yr^{-2}$. A small inflection of the iso-uncertainty lines is observed over the TP period that is due to the large

uncertainties associated to this mission in our uncertainty budget (see Sect. 2.3), as for the GMSL trend uncertainties.



**Figure 4.** Comparison of the uncertainty envelops ($1.65\sigma$) between the L2P 21 and L2P 18 GMSL records. The values shown correspond to the half-width of the confidence interval. The vertical lines indicate the switches of mission.

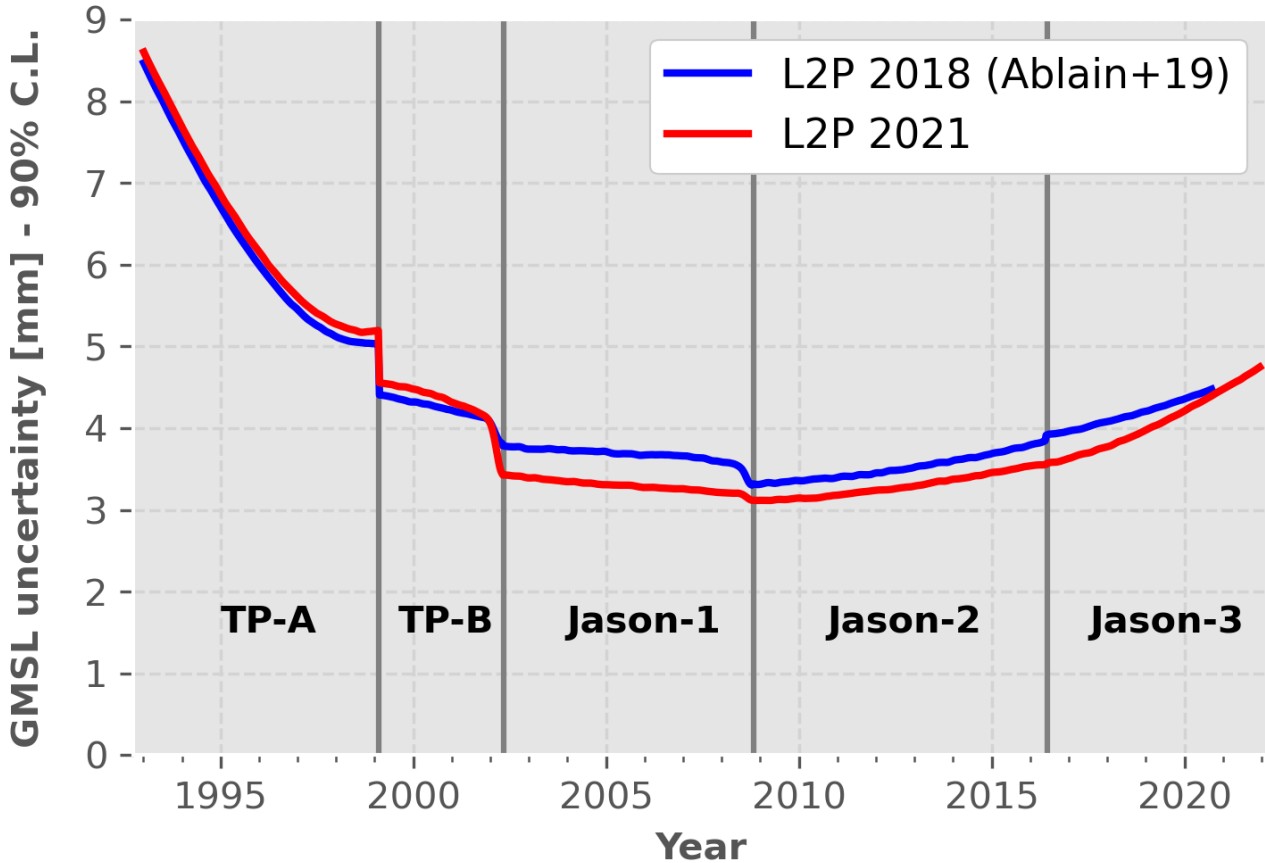

## 4 Contributors to the GMSL uncertainties

Accuracy and stability requirements of the GMSL record have been stated in the literature to allow scientific questions on climate change to be answered. The two intergovernmental organisations, the Global Climate Observing System (GCOS) and the Intergovernmental Panel on Climate Change (IPCC) have published their recommendation about the GMSL trend uncertainty stability: $0.3\ mm.yr^{-1}$ (90 % C. L. ) over 10-year periods. Based on the analysis of the GMSL trend uncertainty presented in Sect. 3.2, we find that the L2P 21 CNES/AVISO GMSL record does not meet the requirement. To our knowledge, none of the GMSL records distributed in the literature meet this requirement.

To identify the limiting factors to the GMSL monitoring stability highlighted above, we here investigate the relative contribution of each uncertainty budget contributor to the total GMSL uncertainty budget. The aim is to identify the main contributors




**Figure 5.** GMSL trends (left) and trend uncertainties (right) of the L2P 21 product estimated for all periods length between 5 years and the total altimetry record (January 1993 to October 2021). Only the significant values (i.e., above uncertainties) are shown. The x-axis corresponds to the central year of the considered period (y-axis). The uncertainties are shown at the 90% confidence level.

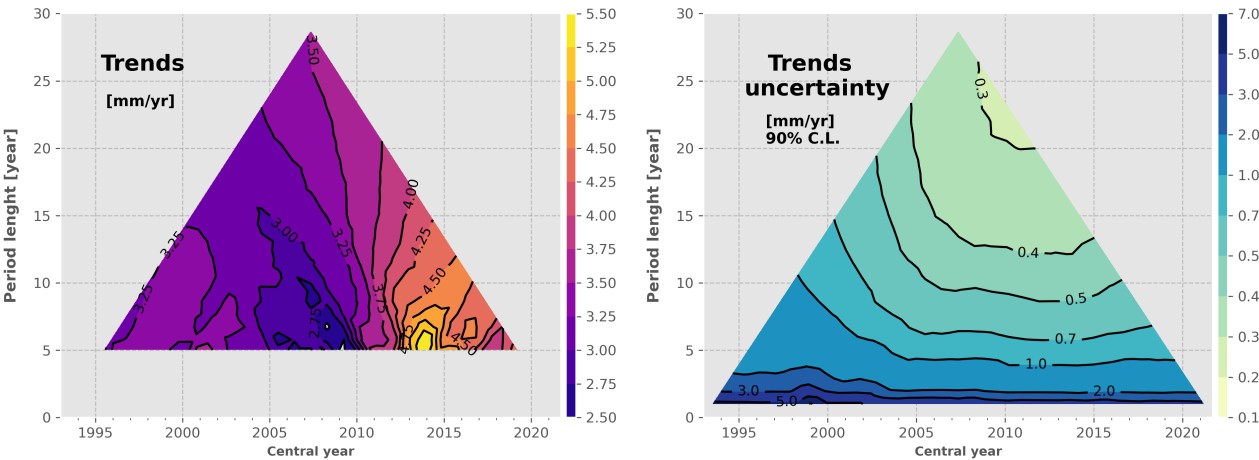

**Figure 6.** GMSL accelerations (left) and acceleration uncertainties (right) of the L2P 21 product estimated for all periods length between 10 years and the total altimetry record (January 1993 to October 2021). Only the significant values (i.e., above uncertainties) are shown. The x-axis corresponds to the central year of the considered period (y-axis). The uncertainties are shown at the 90% confidence level.

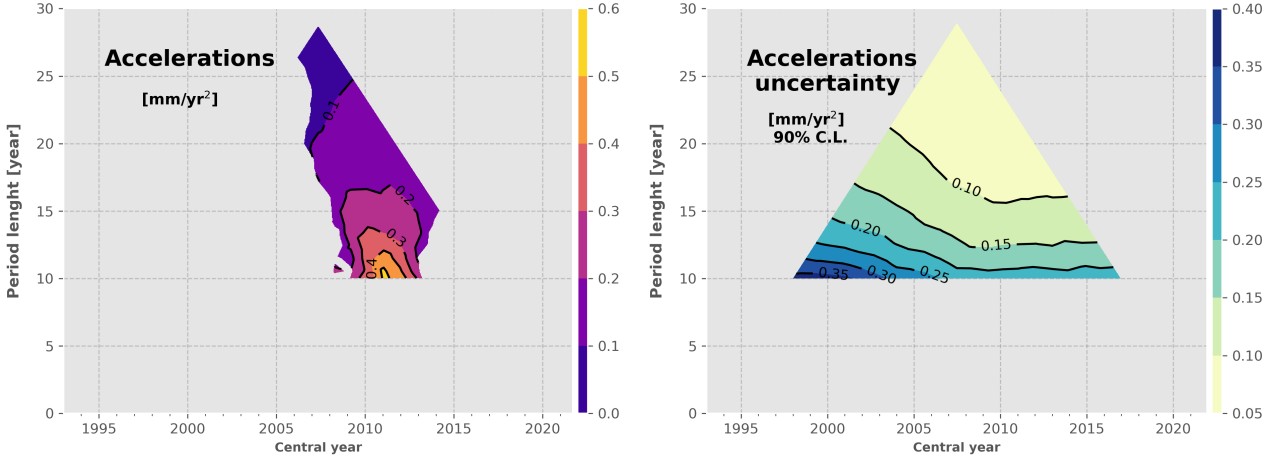

and thus being able to suggest key topics of investigations to tame down the uncertainty and get closer to the stability requirements.

To do so, we derive for each contributors to the GMSL uncertainty budget presented in Sect. 2.3, the GMSL trend variance induced by the contributor on its own (i.e., as if it would be the only source of uncertainty). Since we assumed that each sources of uncertainty are independent from one another, the total GMSL trend variance is the sum of each variance contributor. As a





consequence, one can obtain the contribution (in percent) of each source of uncertainty to the total GMSL trend uncertainty by simply dividing the GMSL trend variance of the respective contributor by the total GMSL trend variance. We perform this operation for each contributor listed in Table 4 and for all periods of the uncertainty diagram. The results are shown in details in Appendix A, and we here focus on the 10-years periods (Figure 7) and 20-years periods (Figure 8).

Figure 7 shows that the trend requirement ($0.3\ mm.yr^{-1}$) is not achieved for any of the 10-year periods over the altimetry

record, the lowest trend uncertainty being $0.45\ mm.yr^{-1}$ for periods centered around 2012. In the first years of the record, the main contributors to this uncertainty level (bottom panel) are the TP drift uncertainties (from 50 % and decreasing) as well as the intermissions offset between TP-A and TP-B (reaching up to 30 % in 2001). For 10-year periods centered after 2004, the latter two sources contribute to less than a few percents and three other sources start to account for more than 80 % of the GMSL trend uncertainty: the radiometer WTC and the correlated errors at 2-months and 1-year. The correlated errors at

2-months and 1-year are derived from the variance of the total sea level signal at 2-months and 1-year assuming that at these time scales the signal can be overwhelmed by noise. As such, the exact sources that create the uncertainties at 2-months and 1-year are unknown. Given our conservative method to estimate them they likely include some actual geophysical signals, and thus, they are likely overestimated. For periods centered after 2008, the radiometer WTC and the uncertainties correlated at 1-year become the major contributors (30-40 % each) whereas the contribution of the 2-months correlated errors decrease

to $\sim 10$ %. Regarding the systematic drift uncertainties: the GIA uncertainty does not contribute to more than 5 % over the full GMSL record, whereas the ITRF uncertainty contributes up to $\sim 15$ % at the end of the altimetry record, as much as the 2-months correlated errors. Finally, the orbit determination uncertainties contribute to less than 5 % and is thus not a major contributor to the sea level trend estimates at the global scale.

Figure 8 shows similar analysis for periods of 20-years. We observe that the lowest trend uncertainty achieved for this

period length is $0.3\ mm.yr^{-1}$ from 2009-present close to the 10-years requirement. This is because at 20-year time scales most of the time-correlated noise has vanished. As shown in the bottom panel, we find that the two main contributors to the GMSL trend uncertainty over periods of 20-years are: the radiometer WTC and the ITRF realisation, from 20 - 30 % for periods centered after 2008. About the other contributors, the two type of correlated errors contribute between 10 - 20 % whereas the uncertainties affecting the TP missions are first, dominant for 20-years periods centered between 2003-2005 and then, rapidly

decrease down to a few percents. Note that the high uncertainty on the intermission offset between TP-A and TP-B ($2\ mm$ in 1999, see Table 4) contributes to less than 10 % only for 20-years periods centered after 2007. It highlights the fact that poor quality tandem phases and/or the lack of it impacts the stability of the GMSL record over long periods. Finally, the uncertainties on the GIA and on the orbits determination are not contributing to more than 10 % for any periods.

Figure 9 and 10 show the same analysis but for the uncertainties of the GMSL acceleration, for periods of 10 and 20

years, respectively. We find that the 1-year correlated errors are the dominant source of uncertainties (40-70%) for the GMSL acceleration over any 10-year periods. Over 20-year periods, the uncertainties on the TP mission are the major contributors before 2006 (60-30%) and the WTC becomes dominant (30-50%) after 2006, mostly due to the related uncertainty increase over the Jason-3 period. Again, it is interesting to note that the large uncertainties on the GMSL offset between TP-A and TP-B contributes significantly to the GMSL acceleration uncertainties. The ITRF and GIA are not contributing to the GMSL





**Figure 7.** Relative contribution of each uncertainty budget contributors to the GMSL trend uncertainty over periods of 10 years. (Top) GMSL trend uncertainty of the L2P 21 record. The brown horizontal dashed line shows the GCOS/IPCC requirement for the GMSL trend stability, i.e., $0.3 \ mm.yr^{-1}$ (90% C.L.). (Bottom) Relative contribution to the total GMSL trend uncertainty (in %) of each uncertainty budget contributors, i.e., the altimeters, the radiometers, the orbit determination and the geophysical corrections.

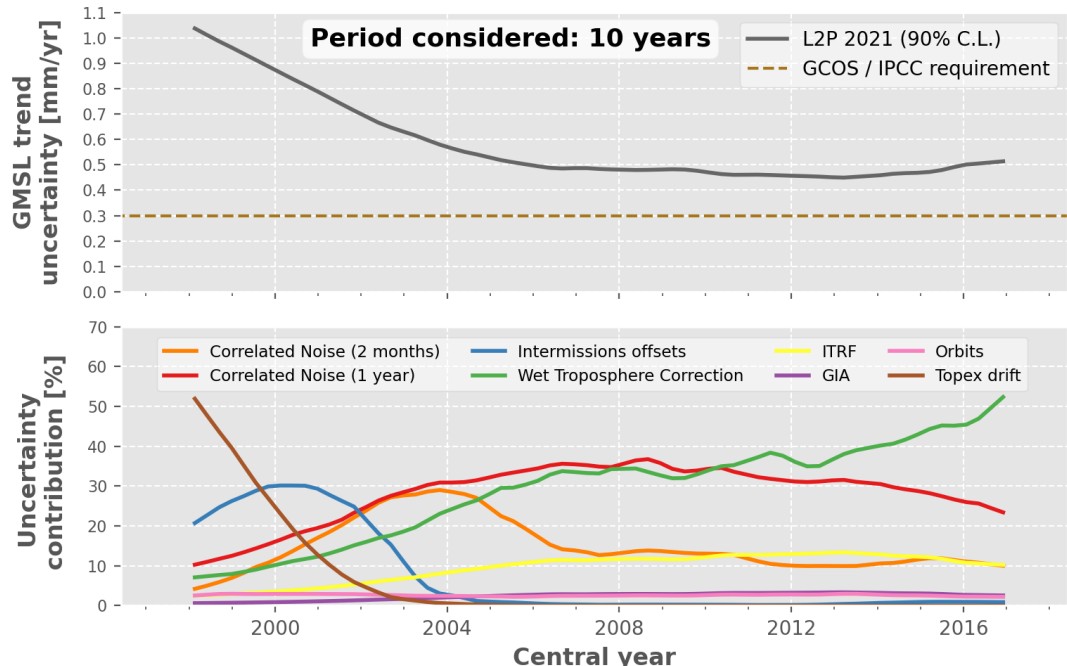

acceleration uncertainty over any periods. This is logical as they are constant drift uncertainties over the full altimetry era, thus not impacting the acceleration (in contrast, the TP altimeter drift uncertainty has an impact since it covers only a fraction of the altimetry era). This is also the case for the orbit uncertainties, however, the latter are small as compared to the other type of uncertainties and do not contribute to more than 5% to the GMSL acceleration uncertainty over any periods.

## 5 Discussion

As detailed in Section 4, we have found four major contributors to the GMSL trend and acceleration uncertainties: the correlated errors at short timescales (2-months and 1-year), the WTC from radiometers, the Topex-Poseïdon data quality and the ITRF realisations. In this section we discuss some implications of reducing such uncertainties to meet the GMSL stability requirements stated by the GCOS.

At the beginning of the altimetry era, our analysis shows that significant improvements need to be achieved on the TP data quality. A new reprocessed dataset in GDR-F standards is on-going and should be publicly released soon. Significant




**Figure 8.** Relative contribution of each uncertainty budget contributors to the GMSL trend uncertainty over periods of 20 years. (Top) GMSL trend uncertainty of the L2P 21 record. The brown horizontal dashed line shows the GCOS/IPCC requirement for the GMSL trend stability, i.e., $0.1\ mm.yr^{-1}$ (90% C.L.). (Bottom) Relative contribution to the total GMSL trend uncertainty (in %) of each uncertainty budget contributors, i.e., the altimeters, the radiometers, the orbit determination and the geophysical corrections.

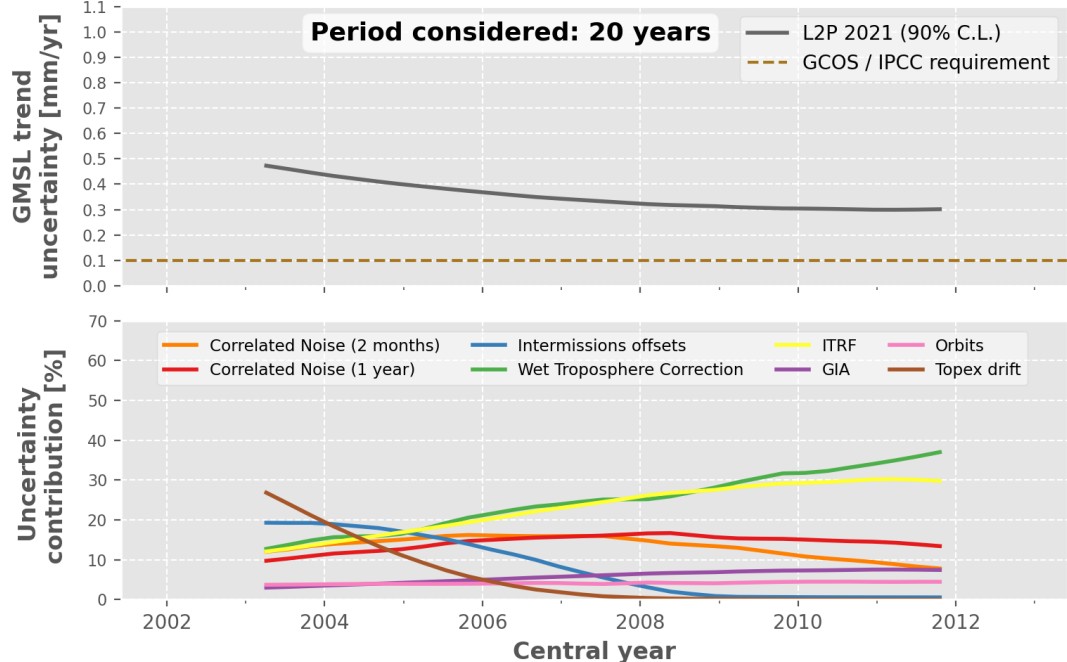

improvements are expected on the stability of the TP-A altimeter as well as on the offset estimation between the two altimeters. Despite the high expectation of the community for such reprocessing, the stability performances of the resulting GMSL will still not be better than it currently is with the last altimetry missions such as Jason-3. Indeed, the other three main uncertainty

contributors will still limit the GMSL stability to about $\pm 0.5\ mm/yr$ over 10-years period. More fundamental improvements are thus needed. Based on our analysis, these are of two types: (i) a better characterisation and understanding of the sources of uncertainties at short timescales (annual and below) and (ii) strong innovations on the current altimetry observing system, on the on-board radiometers and better ITRF realisations stability.

      As we note in Section 2.3, the origins of the short timescales uncertainties affecting the GMSL record are currently unknown.

To overtake this issue, they are empirically estimated and thus have mixed origins, i.e., altimeter noise, radiometer noise, geophysical corrections uncertainties, and more problematic, includes some geophysical signals. As of today, this is a limitation of our uncertainty budget description. A new ESA project named ASELSU (Assessment Sea Level Rise Stability Uncertainty) is currently addressing this limitation by characterising and quantifying the stability uncertainties of the Sentinel-6A Michaël Frielich mission with a careful and exhaustive propagation of the instrumental noises to the system uncertainty budget. Such

project aims to give a comprehensive description of the uncertainties of the altimeter instrument, especially, from Level-0





**Figure 9.** Relative contribution of each uncertainty budget contributors to the GMSL acceleration uncertainty over periods of 10 years. (Top) GMSL acceleration uncertainty of the L2P 21 record. (Bottom) Relative contribution to the total GMSL acceleration uncertainty (in %) of each uncertainty budget contributors, i.e. , the altimeters, the radiometers, the orbit determination and the geophysical corrections.

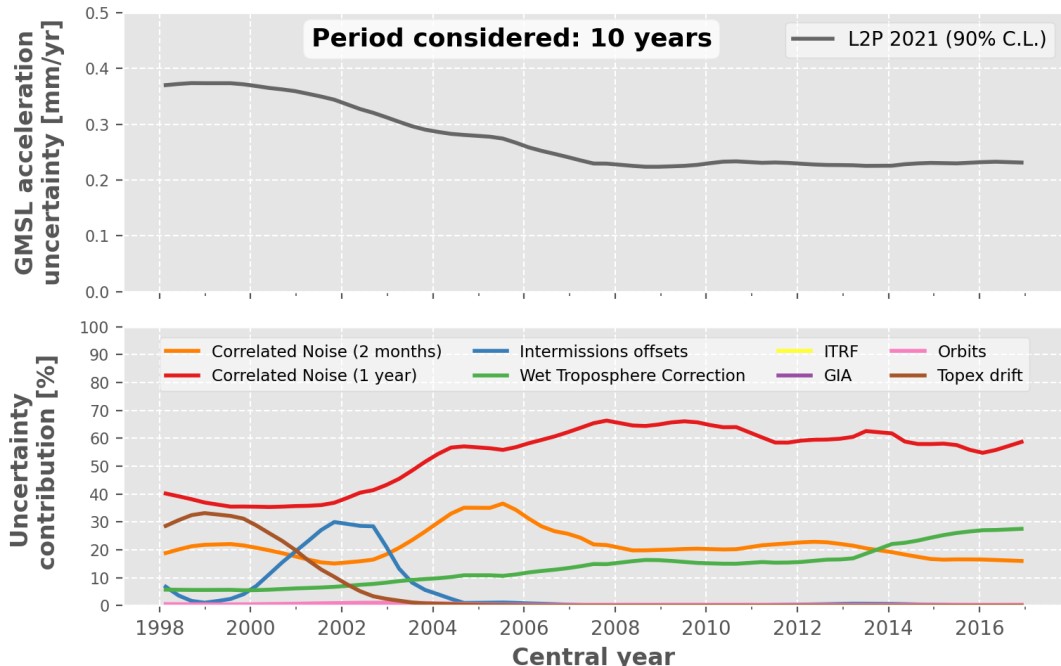

to Level-4 data, based on a metrology approach (Mittaz et al., 2019). Thanks to such work, we will be able to pinpoint the exact origins of the GMSL uncertainties at short timescales and to identify the various improvements needed on the current altimetry observing system to meet the scientific need. Such approach is also used within the FDR4ALT ESA project on the characterisation of the radiometer instruments uncertainties.

Finally, we found that a systematic limit to the GMSL stability appears, which is the realisations of the ITRF. Improvements on the uncertainties of such reference frame represent huge effort from many different scientific communities and governmental organisations. This might be the true limiting factor of the current observing system to the GMSL record stability.

## 6   Conclusions

We have presented the latest release of the CNES/AVISO GMSL record based on the reprocessed CNES L2P 21 1 Hz along-track data of the reference missions, Topex-Poseïdon, Jason-1, Jason-2 and Jason-3. This dataset covers the period January-1993 to December-2021 and it is now provided with an estimate of its uncertainties, available on-line, as well as an empirical correction of the TP-A altimeter drift as proposed in Ablain et al. (2017). The GMSL rise is estimated to $3.3 \pm 0.3 \; mm.yr^{-1}$



**Figure 10.** Relative contribution of each uncertainty budget contributors to the GMSL acceleration uncertainty over periods of 20 years. (Top) GMSL acceleration uncertainty of the L2P 21 record. (Bottom) Relative contribution to the total GMSL acceleration uncertainty (in %) of each uncertainty budget contributors, i.e. , the altimeters, the radiometers, the orbit determination and the geophysical corrections.

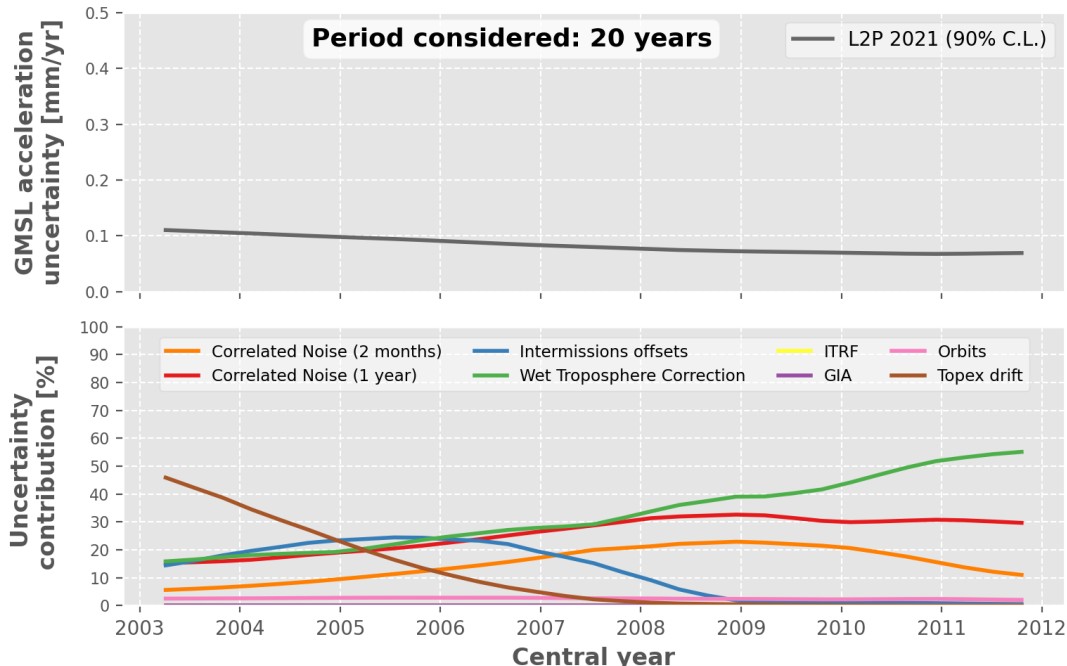

and its acceleration is detected to a rate of $0.12 \pm 0.05\ mm.yr^{-2}$. The GMSL uncertainties, based on an updated version
of its uncertainty budget, are reduced as compared to the previous CNES/AVISO record. This is mostly due to improved instrumental standards and geophysical corrections proposed in the input data products. A few improvements on the method have been presented, such as a new statistical method to estimate the GMSL intermission offsets and its related uncertainties. We showed that the intermission offset uncertainties are reduced when using as many as possible tandem phase measurements. We also updated the uncertainties associated to the WTC of the Jason-3 radiometer that is suspected to show higher instability
than the other radiometers on-board of altimetry missions. This impacts the stability of the GMSL at the end of the data record.

A major result of this paper is the quantification of the respective contribution to the GMSL uncertainties of the individual uncertainty contributors. We have highlighted the results for different time scales and found that the stability of the GMSL record is limited by four major contributors: the correlated errors at short timescales (2-months and 1-year), the WTC from radiometers, the Topex-Poseïdon data quality and the ITRF realisations. Whereas two of these sources of uncertainties are well
identified and will certainly be relatively easily addressed (i.e. , the TP data quality and the WTC stability), the two others clearly set the current limitations of the altimetry observing system (i.e. , the ITRF realisations) as well as of our knowledge on the description of its uncertainties (i.e. , mixed origins in the description of the annual correlated uncertainties). Our results





challenges the altimetry observing system as it is designed today and highlights clear topics of research to be explored in the future to help the altimetry community to improve the GMSL accuracy and stability.




**Appendix A: Variance contributions to the GMSL trend**

We here present the relative contribution to the GMSL trend variance of each uncertainty budget contributors. The uncertainty contributors are detailed in Table 4 and the way to derive their variance contributions is explained in Sect. 4.

- **High-frequency correlated errors (2-months):** These errors contribute mainly to period length shorter than 2-years at a level larger than 40 % of the total GMSL trend variance. It contributes to 10 % and less for periods longer than 10 years
with a peak around 2002 that corresponds to the TP-A/-B switch.

- **High-frequency correlated errors (1-year):** This error contributes mainly to period length between 2-7 years at a level of 40-60 % of the total GMSL trend variance. It contributes less than 30 % for periods longer than 10 year. The smaller contribution at the beginning of the altimetry era (1993-1999) is due to the presence of the additional TP uncertainties.

- **Wet Troposphere Correction from radiometer:** These uncertainties contribute mainly to period longer than 5-years
centered at the end of the altimetry record (after 2007), at a level of 30-40 % of the total GMSL trend variance. The shape of the contribution onto the uncertainty tree is mainly due the occurrence of the other sources of errors, i.e., the TP drift before 2002 and the high-frequency correlated noise for any periods below 5-8 years. The small increase to 50 % in 2016 for periods of 10 years is due to the larger uncertainties that affect J3 WTC.

- **Large frequency errors from the orbit solutions:** This source of uncertainty contributes to less than 5 % for all period
length over the full altimetry record. This result confirms the fact that orbit errors are not contributing significantly at the global scale, as compared to other sources of uncertainties. At the regional scale, it is a major contributor (Prandi et al., 2021).

- **Intermission offsets**: The intermisson offset uncertainty between TP-A and TP-B is responsible for 30-40 % of the GMSL trend variance for periods of 5-10 years centered in the years 1999. The lower uncertainties associated to the other
intermissions offset contribute to less than 5 % of the GMSL trend varaince. This plot illustrates well the advantages to perform tandem phases between altimetry missions.

- **Topex-Poseïdon drift:** This source of uncertainty contributes to 40-60 % of total GMSL trend variance at the beginning of the altimetry time series (before 2000) for periods longer than 5-years. Its contribution follow the left-edge of the uncertainty tree and contributes by a large factor ($\sim$20 %) even for 20 years long and longer periods. It highlights the
need for improved TP data quality to reduce the GMSL trend stability unciertianty.

- **ITRF drift uncertainty**: This source of uncertainty is a constant drift ($0.1 mm.yr^{-1}$) over the full altimetry era. However, its relative contribution is the most important for period length longer than 15 years, centered towards the end of the time series ($> 20$ %). This increase is due to the other contributors' decrease in this part of the diagram. This results highlights a strong limitation to achieve the GMSL stability requirement.




– **GIA drift uncertainty**: This source of uncertainty contributes to 5 % and less for all period considered within the
altimetry records. Its impact is negligible at the global scale whereas it is a major contributor at the regional scale (Prandi
et al., 2021).

**Figure A1.** Variance contribution to the GMSL trend variance of the high-frequency errors correlated at 2-months.





**Figure A2.** Variance contribution to the GMSL trend variance of the high-frequency errors correlated at 1-year.



**Figure A3.** Variance contribution to the GMSL trend variance of the Wet Troposphere Correction from radiometer.







**Figure A4.** Variance contribution to the GMSL trend variance of the orbit solutions.







**Figure A5.** Variance contribution to the GMSL trend variance of the intermission offsets.



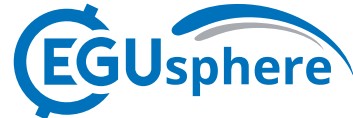

**Figure A6.** Variance contribution to the GMSL trend variance of TP data quality.





**Figure A7.** Variance contribution to the GMSL trend variance of the ITRF drift uncertainty.




**Figure A8.** Variance contribution to the GMSL trend variance of the GIA drift uncertainty.





## Appendix B: Variance contributions to the GMSL acceleration

We here present the relative contribution to the GMSL acceleration variance of each uncertainty budget contributors. The
uncertainty contributors are detailed in Table 4 and the way to derive their variance contributions is explained in Sect. 4.

- **High-frequency correlated errors (2-months)**: These errors contribute mainly to period length shorter than 5-years at
  a level larger than 50 % of the total GMSL trend variance. It still contributes to 10-20 % for periods longer than 10 years.

- **High-frequency correlated errors (1-year):** These errors contribute to almost any periods over the altimetry era to a
  level of 30-60 % of the total GMSL acceleration variance. It is clearly the largest contributor to the GMSL acceleration
  variance over all other sources of uncertainties.

- **Wet Troposphere Correction from radiometers:** These uncertainties contribute mainly to period longer than 10-years
  centered at the end of the altimetry record (after 2010), at a level of 30-50 % of the total GMSL acceleration variance.
  The increase towards periods on the right-edge of the "uncertainty tree" is due to the larger uncertainties that affect J3
  WTC.

- **Large frequency errors from the orbit solutions:** This source of uncertainty does not contributes to the GMSL accel-
  eration variance at the global scale over the full altimetry record. At the regional scale, it is a major contributor (Prandi
  et al., 2021).

- **Intermission offsets**: The intermisson offset uncertainty between TP-A and TP-B is responsible for 10-20 % of the
  GMSL acceleration variance over periods that covers the TP mission. The lower uncertainties associated to the other
  intermissions offset contribute to less than 5 % of the GMSL acceleration variance.

- **Topex-Poseïdon drift uncertainty:** This source of uncertainty contributes to the GMSL acceleration variance up to 30-
  50 % over the left-edge of the "uncertainty tree". This is due to a change in the drift uncertainty values prescribed to the
  two altimeters of TP. This is this change of slope, i.e., that can be assimilated as acceleration that makes the contribution
  high.

- **ITRF drift uncertainty**: This source of uncertainty does not contribute to the GMSL acceleration variance. This is
  expected since the associated uncertainty is a constant drift over the full altimetry era.

- **GIA drift uncertainty**: This source of uncertainty does not contribute to the GMSL acceleration variance. This is
  expected since the associated uncertainty is a constant drift over the full altimetry era.

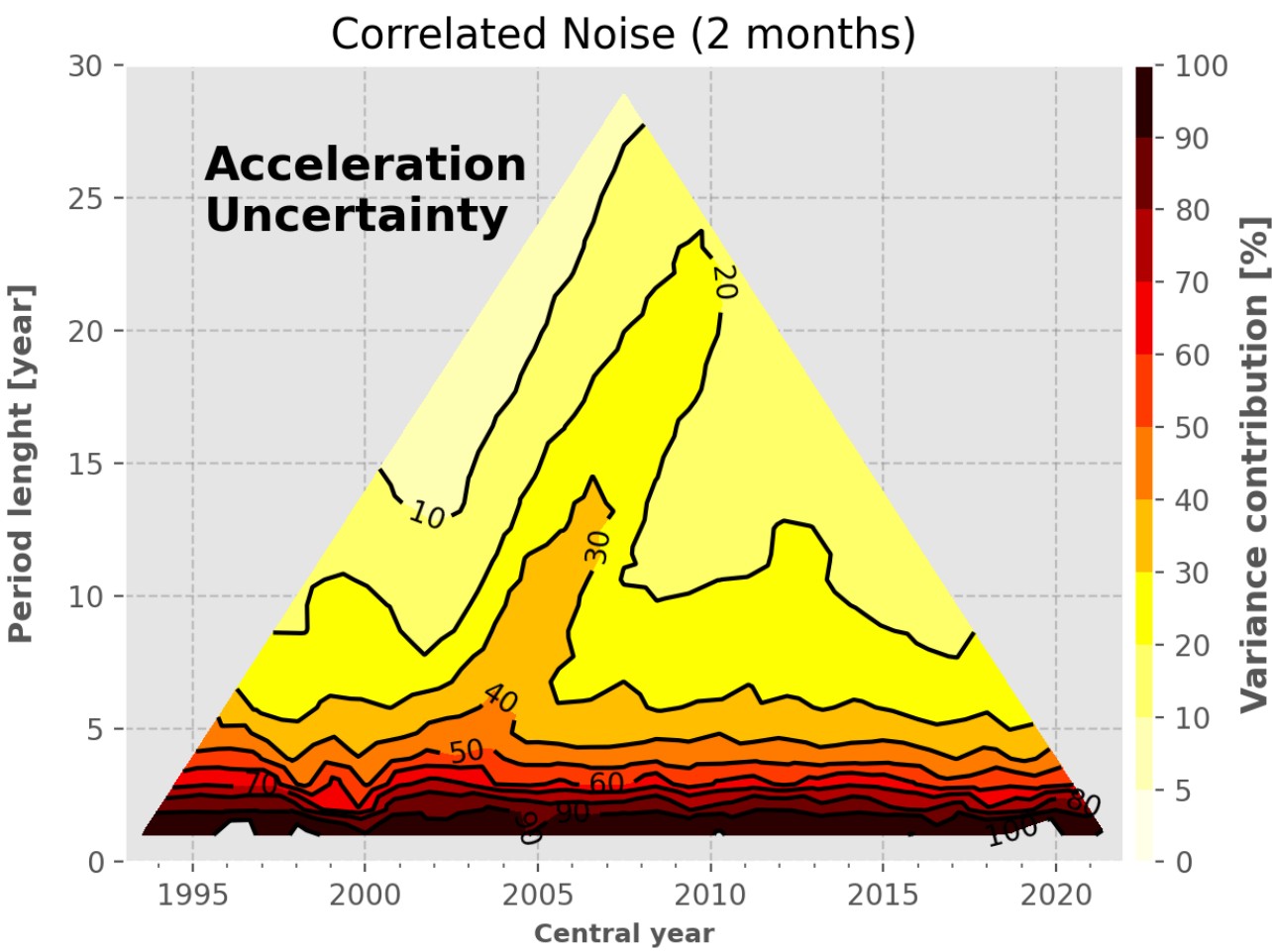

**Figure B1.** Variance contribution to the GMSL acceleration variance of the high-frequency errors correlated at 2-months.





**Figure B2.** Variance contribution to the GMSL acceleration variance of the high-frequency errors correlated at 1-year.





**Figure B3.** Variance contribution to the GMSL acceleration variance of the Wet Troposphere Correction from radiometer.



**Figure B4.** Variance contribution to the GMSL acceleration variance of the orbit solutions.







**Figure B5.** Variance contribution to the GMSL acceleration variance of the intermission offsets.





**Figure B6.** Variance contribution to the GMSL acceleration variance of TP data quality.





**Figure B7.** Variance contribution to the GMSL acceleration variance of the ITRF drift uncertainty.


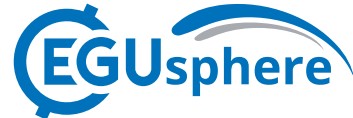

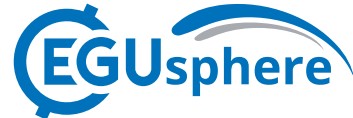

**Figure B8.** Variance contribution to the GMSL acceleration variance of the GIA drift uncertainty.



*Data availability.*    The GMSL dataset described in this manuscript is available to download at the following address: https://www.aviso.altimetry.fr/en/data

410    indicators-products/mean-sea-level/data-acces.html .

*Author contributions.*    AG led the study, performed the computations and wrote the manuscript with some inputs from the authors. BM wrote the introduction of the manuscript and contributed to the discussion. AR gave strong support to established the statistical approach used in Sect. 2.2.1. MA and PP discussed the results and helped improving the key messages. All authors contributed to revised the manuscript.

*Competing interests.*    The authors declare no competing interests.

415    *Acknowledgements.*    This work has been supported by the CNES French space agency in the framework of the SALP project.



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
