# Peer review of "Current observed global mean sea level rise and acceleration estimated from satellite altimetry and the associated measurement uncertainty"

_EGUsphere, 2022_

## Referee Comment (RC2)

I am one of the reviewers for this study.

The current study is a replication of a previous study by Ablain et al., (2019) using additional satellite altimetry (SA) data acquired since 2017.  Both studies first construct a full variance-covariance (V/C) matrix of SA measurement errors.  They then use a quadratic model to represent the systematic components of the observed GMSL anomalies.  They identify coefficients of the quadratic as trend and acceleration during the SA period (1993-2022).  Both studies used the ordinary least squares (OLS) to estimate the parameters of the quadratic representation.  An elaborate scheme is used to assess the omission of the V/C matrix in the OLS solution on the uncertainties of the estimated trend and acceleration.

1 – If a proper V/C matrix is constructed, the simplest approach to quantify its impact on the model parameters would be to carry out a generalized least squares (GLS) solution, which makes use of the V/C matrix together with the OLS solution.   Various solution statistics can then be compared to assess the impact of the V/C matrix on the adjusted GMSL anomalies and the model parameters.  I wonder why the investigators have decided to use a convoluted scheme and ignored this obvious and direct assessment?

2 – Both studies state that the estimates for the parameters are unbiased if OLS is used.  The unbiasedness of the estimated parameters using OLS as well as GLS are both conditional on the "expected values" of the disturbances being zero.  However, this assumption cannot be established *a priori* since the purpose of modelling SA born measurement errors after all is to construct a realistic V/C matrix to achieve this end.

3 – An implicit assumption was also made that the quadratic model captures all the systematic variations in the observed GMSL anomalies.  Again, there is no *a priori* evidence to support this assumption (I will discuss later that this is a faulty premise).    Hence, the residuals, as estimates of the random disturbances, may biased unless shown otherwise.

4 – The unbiasedness assumption of the estimated quadratic model parameters using OLS or GLS does not guarantee that these estimates are comparable.  If some of the GMSL anomalies with high leverages are not of good quality, they will *bias* the quadratic model parameters using OLS.  These low-quality high leverage data will not impact the estimates if the GLS is used

provided that their uncertainties are properly reflected in the V/C matrix (i.e., they are down weighted). Consequently, assuming equivalency between the estimated parameters using OLS and GLS is not always warranted with certainty in practice. Therefore, using estimated parameters via OLS solution for the assessment of the V/C matrix may be faulty.

5 – If the unbiasedness of the estimated parameters is demonstrated to be valid despite the pitfalls discussed earlier, their uncertainties are still *biased* if OLS is used because *the standard error (SE) of the solution – the a priori variance of unit weight ~weighted root mean square error of the residuals* is biased (Toutenburg 1984, eqn. 2.2.45 pg. 31). There is no mention about this bias in both studies, whether it was properly accounted for in quantifying the uncertainties of the model parameters and its magnitude. This issue is important because both studies are about the uncertainties. Consequently, it is likely that the major syntheses of this study *"the stability performances of ±0.3mm/yr at the 90% confidence level (C. L.) for its trend and ±0.05mm/yr2 for its acceleration over the 29-years of the altimetry record"* were mis quantified.

6 – Another problem is modelling the effect first order serial correlation of the disturbances, AR(1) of the GMSL anomalies during 1993 – 2022. This effect is different than the bias in the SE caused using OLS discussed previously. Their presence in the observed GMSL anomalies is not recognized in both studies. Modelling the full V/C matrix does not make this effect to disappear. GMSL anomies averaged yearly, monthly, 10-day intervals (and smoothed) exhibit extremely high serial correlation with correlation coefficient, as high as 0.94 (Iz and Shum, 2020). Similar effects are also observed frequently in tide gauge records that may have AR(1) correlation coefficient as large as 0.7. Ignoring their presence will lead Type II errors in null hypothesis testing the statistical significance of the estimated model parameters. Only a handful of past studies on sea level have recognized this phenomenon. In the case of GMSL, the study by Nerem et al. (2018) corrected its effect by discounting the pertinent uncertainties using a relationship in their study (surprisingly, the magnitude of the estimated AR(1) correlation coefficient was not published!). Note that the formula for the *sample reduction* coefficient used in their study was derived using a relationship derived for univariate cases (WMO, 1966) and not necessarily valid for multiple regressions (luckily the outcome was not fatal). In any case, there

is no indication if the V/C matrix of the GMSL records incorporated the AR(1) effect in this manuscript (I am not referencing to the analyses of their tandem phases in the manuscript).

7 – Proper statistical formalism demands that the AR(1) effect in the GMSL records to be carefully analyzed and accounted for in the V/C matrix of the observations if it is statistically significant. Previous few studies used the simplistic sample reduction approach, which assumes that the GMSL disturbances are identically and independently distributed *i.i.d*, which is false now as evidenced by this study's *full* V/C matrix. Consequently, the V/C matrix of this study requires a construct akin to a *heterogeneously first order autocorrelated* V/C matrix, abbreviated as ARH(1) or ARW(1) for the V/C matrix.

8 – The use of quadratic model to represent the systematic components of the SL and GMSL anomalies without identifying its underlying physics unambiguously is another important source of error. The non-linear coefficient of the quadratic is stated as twice the *acceleration*, and the coefficient of the linear parameter as *trend*. Both terms are colloquial and vague in the context of sea level studies with important ramifications in describing the kinematics of SL anomalies. The acceleration is a "*uniform acceleration*" with implications. The acceleration estimated through a quadratic model must be the same (uniform) for various lengths of the records. If not, then the acceleration is not uniform. It can be transient, episodic, or a variable acceleration with different physical attributes for the underlying phenomena occurring throughout the records. If the uniformity of the estimated acceleration is not demonstrated, then the model and the findings will be questionable.

9 – A statistically significant uniform acceleration demonstrably present in the GMSL anomalies will change the meaning of the *trend* parameter. In this case, the estimated coefficient of the linear component of the quadratic model is "*the initial velocity.*" *This estimate varies depending on the choice of the initial epoch of the quadratic model.* Hence, by choosing different epochs for the time variable, one will get different estimates for the so-called *trend*. There is no indication that this fact was recognized not only in this study but also in others resulting in different estimates for the initial velocities labeled as invariant trends generating a plethora of

differing values.  The casual use of the term *trend* in this study in the presence of a likely uniform acceleration is therefore misleading.

10 – The underlying kinematic meaning of the quadratic model opens another source of *ambiguity.*  The acceleration component of a kinematic model can also be represented equally well with a low frequency variation in GMSL (Iz and Shum, 2020).  Such periodicities were already detected in TG records.  A visual examination the plot of the filtered GMSL anomalies suggests that there is steady trend in GMSL rise until 2011 (w/o acceleration) followed by a change point during 2011 with a mean shift followed by another trend (note that I am not promoting my visual examination as a reliable scientific tool).  If verified, another possibility is a broken trend model (Iz et al., 218) leading to an *average acceleration* in GMSL with an underlying causality to be investigated.  Why does this issue matter?  A kinematic model with a periodicity or with a uniform acceleration, or average acceleration all will predict similar GMSL anomalies for the near future.  Yet their differences will diverge markedly for predictions extending several decades.   This would be an extremely costly blunder in climate mitigation decision making.

11 – The other issue affecting the current study and its model for predictions involves using *smoothed* GMSL records for the analyses.  Smoothed records will result in loss of GMSL variability information in predicting future anomalies rendering the usefulness of the predicted anomalies.  The AR(1) effect in smoothed records will also increase.

12 – Statistically significant signature of periodicities of luni-solar origin have been detected in various TG records.  Recently, Iz (2022) also demonstrated their presence in GMSL anomalies in 120 yearlong yearly averaged records of the GMSL budget components compiled by Frederikse et. al., (2020).   The signature of the luni-solar forcings also present in the series generated by this study.  They are statistically significant ($\alpha$=0.05).  Their presence suggests an incomplete quadratic model resulting in non-random residuals, reinforcing the earlier criticisms about the assumptions of the methodology (the expected values of the disturbances being zero) used in this study for assessing the impact of the V/C matrix.

The efforts to build a full V/C matrix is a worthy endeavor for optimal analyses of GMSL anomalies. However, I found the outcome of this study dubious due to the outlined methodological flaws. The estimated model uncertainties are likely to be biased, there is no evidence if the constructed V/C matrix of the disturbances accounts for the AR(1) effect, the kinematic representation of the GMSL rise through a quadratic model is ambiguously presented, incomplete, and misleading in its interpretation.

H. Baki Iz

7/19/2022

Ablain M., Meyssignac B., Zawadzki L., Jugier R., Ribes A., Spada G., Benveniste J., Cazenave A., and Picot N., 2019, Uncertainty in satellite estimate of global mean sea level changes, trend and acceleration, Earth Syst. Sci. Data, 11, 1189–1202.

Frederikse, T., F. Landerer, L., Caron, S. Adhikari, D. Parkes, V.W. Humphrey, S. Dangendorf, P. Hogarth, L. Zanna, L. Cheng and Y.H. Wu, 2020, The causes of sea-level rise since 1900, Nature 584, 393–397.

İz H.B., 2022: Low frequency fluctuations in the yearly misclosures of the global mean sea level budget during 1900 – 2018, to be published in J. Geod. Sci.

Iz H.B., C.K. Shum, 2020: Certitude of a global sea level acceleration during the satellite altimeter era, J. Geod. Sci., Vol. 10, pp. 29–40.

İz H.B., C.K. Shum, C.Y. Kuo, 2018: Sea Level Accelerations at Globally Distributed Tide Gauge Stations During the Satellite Altimetry Era, J. Geod. Sci. Vol. 8, pp. 130–135.

Nerem R. S., B. D. Beckley, J. T. Fasullo, B. D. Hamlington, D. Masters and G. T. Mitchum, 2018, Climate-change–driven accelerated sea-level rise detected in the altimeter era. PNAS, 1-4.

Toutenburg, H., 1982, Prior information in linear models, John Wile and Sons, New York.

WMO, 1966, WMO Technical Note 79. WMO No. 195, TP-10, Geneva.

---

## Author Response (AR1)

**Response to referee #1**

Dear Thomas,

Please find below our complete answer to your comments.

**General**

We have included in the revised version of the paper the comparison of our GMSL record to the GMSL curves you mentioned. See paragraph l.238 and associated figure 5.

**Line-by-Line**

**L.3 /** The syntax of Poseidon has been modified accordingly

**L.22-23 /** We modified the sentence to make the distinction between sampling and accuracy. We also add references to the accuracy numbers. See Lines 23-25.

**L.72 and 79 /** Modifications as follow has been added: " ... all grid cells within +/-66 degrees N/S (the Topex and Jasons coverage) are spatially averaged..." See L.82 now.

**L.134 /** We modified the sentence according to your comment, see l.145-148

**L. 178 /** We estimated the trend and acceleration over the 29 years period of the GMSL record, with and without filtering, and the results are identical. This was expected since the 2 months cut off period of the filter is low as compared to the total lenght of the record (i.e, 29 years). The border effects are thus not significant. This is also true for estimations over 5 years periods.

We apply such a fitlering on the AVISO GMSL record as we consider that we remove some high frequency noise and that we still do not degrade the trend and acceleration estimations. We nevertheless note that raw GMSL time series could be publicly provided. This will be done in a future release.

Figure 5 and 6 / Thank you for having noticed this point. We were using the wrong GMSL timeseries for Figure 5 and 6 (i.e., not corrected for the Topex-A drift). We now obtain, naturally, consistent values between Figure 3, 5 and 6.

**L.294 /** The ITRF uncertainty is certainly non-linear, this is a good point. We modified l.324 accordingly. We used the uncertainties published for the ITRF2014, indeed. The

updated reference frame ITRF2020 should help reducing the associated uncertainties as: time series are longer, seasonal signals are now considered in the local movements of the ITRF2020, the models are enriched as well as more data is used to constraints the model (I.e., Galileo). Information has been added to the manuscript L.358.

**L.409 /** We are currently discussing publishing the scripts used to calculate the trend, acceleration and uncertainties. Unfortunately, it will take some time. In the meantime, Prandi et al. (2021) made public similar scripts to perform OLS estimation with uncertainties in the context of regional MSL. This code is based on the same theoretical approach as ours and can be used to reproduce our analysis. We added this information l.207.

**Response to referee #2**

There is here, from the start, a misunderstanding of the objective of the present study (and of Ablain et al., 2019). This study (and Ablain et al. 2019) does NOT intend to estimate the "systematic components" of the observed GMSL anomalies. We only intend to characterize the uncertainty in GMSL measurements due to the instrumental errors. We state it clearly in the manuscript on line 37 (and also in Ablain et al. 2019 page 1190, 2nd column, 3rd paragraph).

Our group produces sea level measurements from satellite altimetry level 1 data. We are involved in this activity with CNES managers and CNES engineers who developed the radar altimeters onboard the satellite altimeters because we are in charge of the delivery of the sea level scientific product for CNES. We deliver sea level products and associated uncertainties (which is a pure instrumental uncertainty) as a service to the scientific community who can then use these products and their associated uncertainty to evaluate further different elements such as the geostrophic circulation, the mean dynamic topography and its changes,  the GMSL anomalies and, if they wish, the  "systematic components of the observed GMSL anomalies". Here we certainly do not evaluate ourselves the "systematic components of the observed GMSL anomalies", we only provide the updated GMSL anomalies derived from satellite altimetry and its associated instrumental uncertainty. We also provide an estimate of the 1993-2022 trend and acceleration with associated uncertainties as a metric for the low frequency changes in GMSL. The uncertainty on this trend and this acceleration is an uncertainty ONLY due to instrumental errors. We never claim the trend or the acceleration represent any "systematic component" of the GMSL. We provide the trend and the acceleration with uncertainty just as a reference for the scientific community so they can realize the actual amplitude of the instrumental errors on such metrics that are largely used for many different purposes in the science community. Note that our objective is also to provide a reference calculation of the trend uncertainty and the acceleration uncertainty so people can check their own calculation of the instrumental errors on their trend and acceleration estimate when they use our error variance/covariance matrix.

The confusion here is very common in the community that analyses sea level rise. We believe this is because this community is very focused on the detection and attribution of the forced response of global mean sea level to anthropogenic forcing on the climate system. This forced response is expected to take the shape of a parabolic signal on global mean sea level at decadal time scales (according to climate model simulation). For this reason people in this community tend to interpret any parabolic signal in GMSL as a "systematic component" of the GMSL that has some predictive value.

Here we DO NOT do such things. We are addressing a community that is much larger than the single community that analyses sea level rise. We are providing an update of GMSL anomalies with instrumental uncertainties for all the science communities that use GMSL products. These communities range from the Earth water cycle community to the Energy cycle community and it includes many communities as different as the ocean circulation community (which intend to assimilate the GMSL anomalies in ocean models for example) and the geophysics communities (like GIA people or solid earth people who use GMSL anomalies as observational constraint). Many of these communities compute trends or accelerations in their application. For this reason, we compute here one trend and one acceleration (the one trend

and the one acceleration over 1993-2022) with the associated instrumental uncertainty derived from the error variance covariance matrix. This is as a reference so they can get a rapid idea of the actual amplitude of the instrumental error on such metrics. This is also a reference against which they can check their own uncertainty calculation when they use our error variance covariance matrix.

From this review, we suppose that the parabolic signal of the observed GMSL anomalies has been interpreted as "systematic components " of the GMSL. This is something that should NOT be done. A simple trend calculation or a simple acceleration calculation on the measured GMSL time series that we are providing here, does not give an estimate of any "systematic component". There is a misunderstanding of the GMSL data in this interpretation. We are not totally clear on what is meant by the term "systematic component". We suspect he means the forced response of GMSL to the anthropogenic forcing on the climate system. If so, there is actually a long way to isolate the "systematic component" out of the GMSL measurement we are providing here. To isolate this signal, one needs to estimate the internal variability of the climate system and also to estimate the response to other forcings on the climate system such as the sun variability and the long term tides from other planets of the solar system. One also needs to isolate the intrinsic variability generated by the ocean circulation.

We clearly and explicitly explained this point in Ablain et al. 2019 (page 1190, 2nd column, 3rd paragraph). We repeat here in this paper that we are addressing only instrumental uncertainties and we are not trying to isolate the forced response to anthropogenic forcing. Reading your review, we understand that we have not been clear enough in the introduction of this manuscript. We thank you for pointing out this issue unintentionally. We have now added a supplementary paragraph in the introduction to clarify this positioning. See paragraph from line 40.

Because we were not clear enough in the introduction that we are addressing only instrumental uncertainties and do not estimate any "systematic components", we believe that several comments of the review are actually not relevant. We explain this below.

**Comment #1**

We explained in detail in Ablain et al. (2019) why we use an OLS rather than a GLS. OLS estimate is known to be less accurate than GLS in terms of the mean square error, because its variance is larger. A generalized least square estimate would probably help in narrowing slightly the trend uncertainty, but the difference is expected to be small, in particular when the V/C matrix is not far from identity (Ribes et al. 2016). Important advantages of using OLS are that (i) OLS is consistent with previous estimators of GMSL trends as well as estimators of trends in other essential climate variables than GMSL (indeed OLS with V/C matrices is the approach used in the IPCC, see for example Hartmann, et al., 2014,) and that (ii) the OLS best estimate does not depend on the estimated variance–covariance matrix $\Sigma$. Reasons why the OLS estimate could be prefered, even if the V/C matrix is not the identity, were also discussed in the IPCC AR5 (e.g., Chapter 2, Box 2.2)

We tested a GLS estimate on a yearly average GMSL time series (for which the V/C matrix is now invertible) and checked that the result is very close to the OLS estimate. We do find very

similar results with both estimates. See plot below. We added this information in the manuscript now on line 204.

[Figure]

**Comment #2**

There is a confusion here. We do not assume that a linear signal or a parabolic signal represent the GMSL time series and then test these simplistic models against the zero assumption. This approach is used by people trying to isolate the GMSL signal forced by anthropogenic emissions assuming it is a linear trend or a parabolic signal. Here we are not interested in this. Here we intend to provide the most accurate estimate of the observed GMSL. Then we consider the GMSL time series and try to derive the trend and the acceleration of the time series over the record length as metrics of the lowest frequencies included in the time series. In this sense we do not expect the disturbances to be zero and in this sense the OLS estimator itself is an unbiased estimator (see Ribes et al. 2016 and demonstration in the reference therein). Note that linear trend models are also used (and useful) in cases where the underlying shift is not linear in time (and so the "expected value" of the residual is non-zero; see again IPCC AR5, Box 2.2). In such cases, trends are used to measure the rate of increase in a time-series.

**Comment #3**

We never assume the quadratic model captures "the systematic variations in the observed GMSL". This interpretation is biased towards interpreting the GMSL physical signals. Again, we estimate the trend and acceleration of the observed GMSL as metrics of the lowest frequencies included in the time series. See our answer to point #2.

**Comment #4**

We agree with this comment and we refer to our response of point #1 (where we checked that the GLS estimate and the OLS estimate lead to the same estimate) as well as point #2.

**Comment #5**

We would like to make two comments in response to this point. First, the estimated V/C matrix is taken into account in the uncertainty analysis, i.e., quantification of the standard error of the trend and acceleration coefficients. As a result, this uncertainty analysis is valid and reliable – and fully consistent with our V/C estimate. Second, our uncertainty analysis only accounts for the measurement uncertainty – this is now made clearer in the revised version of the manuscript (see our response to reviewer's introduction for more details). In particular, any uncertainty related to internal variability within the climate system (which can be large) is not taken into account here. So, our confidence range is not representative of uncertainty on human-induced SLR.

**Comment #6**

Indeed, if the objective of this study was to isolate the forced response of sea level to anthropogenic emissions we would need to account for the internal variability and the natural variability in GMSL in the least square approach. And the reviewer is right, we would need to model the serial correlation in the GMSL time series. But that is not our objective. Our objective here is only to deliver to the community the most accurate GMSL time series possible from satellite altimetry with associated instrumental uncertainty and to estimate the uncertainty due to instruments on the 1993-2020 trend and acceleration of the GMSL time series. For this reason we believe this comment is not relevant here.

**Comment #7**

See answer of the previous point #6.

**Comment #8**

Of course we expect the forced response of GMSL to anthropogenic forcing to be a response that is more complex than just a parabolic signal. For this reason, the acceleration and the trends are expected to change with time, we agree. But here we do not tackle this problem. We simply want to give metrics for the lowest frequency included in the 1993-2020 GMSL record derived from satellite altimetry. For this reason we focus on one unique trend and one unique acceleration : the 1993-2020 trend and acceleration corresponding to the full satellite altimetry era. Once again, because of the different perspective, we believe the comment here is not relevant.

**Comment #9**

See our answer to the previous comment #8.

**Comment #10**

That is the point here. We use the trend and acceleration as simple metrics for the low frequency and do not make any predictions with it. This would be indeed "an extremely costly blunder in climate mitigation decision making". Any other metric could/should be used for sure to achieve this goal. We choose one metric and are clear about it so people in the community can test their own calculation against ours.

We DO NOT pretend those metrics represent any "systematic component" of the GMSL and we CERTAINLY NOT pretend those metrics have any predictive skills. For your recollection , we are only providing an observed estimate of the GMSL and low frequency metrics with the INSTRUMENTAL uncertainty. We clarified this point in the paper l. 197-198.

**Comment #11**

We do not make any predictive model in this study. See answer to comments #10.

**Comment #12**

Thank you for pointing out that "the efforts to build a full V/C matrix is a worthy endeavor for optimal analyses of GMSL anomalies". We believe so as well. Indeed this is the goal of the paper and we insist on that point: for the altimetry measurements only.

But please note that we DO NOT intend to provide a full V/C matrix that represents the errors of the forced response of GMSL to anthropogenic emissions. Our work is much more modest. We ONLY provide a V/C matrix of INSTRUMENTAL errors in GMSL. Providing a description of the GMSL instrumental errors is a very important step for scientists of all communities that use the GMSL time series and not only for the sea level rise community. For the sea level rise community, our work provides the very first brick over which scientists can further build a complete V/C matrix of the forced signal. This is not our intention. We understand from this review that we were not clear enough. Now we clarify this in a whole new paragraph in the introduction (see lines 40-48). We hope this makes our objective clearer to the sea level rise community.